# Dynamic Boundary of P-Set and Intelligent Acquisition for Two Types of Information Fusion

**Shouwei Li [1]** **, Yao Xiao [1] and Kaiquan Shi [2],***

1    School of Business, Shandong Normal University, Jinan 250014, China; lishouwei@sdnu.edu.cn (S.L.);
     xiaoyaosdnu@stu.sdnu.edu.cn (Y.X.)
2    School of Mathematics, Shandong University, Jinan 250100, China
*    Correspondence: shikq@sdu.edu.cn

**Abstract:** The development of information technology brings the challenge of data redundancy and data shortage to information fusion. Based on the dynamic boundary characteristics of p-set, this paper analyzes the structure and generation of p-augmented matrix, and then analyzes the dynamic generation of information equivalence class, and then proposes an intelligent acquisition algorithm of information equivalence class based on matrix reasoning. In addition, this paper analyzes two types of information fusion, namely information redundancy fusion and information supplement fusion. Then, the relationship among redundant information fusion, supplementary information fusion, and information equivalence classes is analyzed. Finally, this paper presents the application of intelligent acquisition of information equivalence class in information retrieval.

**Keywords:** p-sets; information equivalence class; intelligent acquisition algorithm

## 1. Introduction

Information fusion widely exists in the biological world, and it is an intrinsic feature of organisms from the ancient times to the present [1]. As a hot field of information science, information fusion technology originated from the military application in the 1970s [2]. After the continuous research climax from the early 1980s to now, the theory and technology of information fusion have been further developed rapidly [3,4]. As an independent discipline, information fusion has been successfully applied to military fields such as military command automation, strategic early warning and defense, multi-target tracking, etc., and gradually radiated to many civil fields such as intelligent transportation, remote sensing monitoring, e-commerce, artificial intelligence, wireless communication, industrial process monitoring and fault diagnosis, etc.

Information fusion is a formal framework, which uses mathematical methods and technical tools to synthesize different information, in order to get high-quality and useful information [5–8]. Compared with the single-source independent processing, the advantages of information fusion include: improving detectability and credibility, expanding the space-time sensing range, reducing the degree of reasoning ambiguity, improving the detection accuracy and other performance, increasing the target feature dimension, improving spatial resolution, enhancing the system fault-tolerant ability and white adaptability, so as to improve the whole system performance.

In the past 20 years, scholars have put forward a variety of methods for information fusion, and achieved rich research results [9–12]. Among them, p-set theory and method is a unique application. P-sets (P = packet) is a mathematical model with dynamic boundary features [13–15]. It is obtained by introducing dynamic features into the finite common element set $X$, and improving it. The dynamic boundary features of the p-set are as following: for the given finite set of common elements $X$, and the attribute collection $\alpha$ of $X$, (a) If the attribute $\alpha_i$ is added into $\alpha$, $\alpha$ generates $\alpha^F$, $\alpha \subseteq \alpha^F$, then some

elements are removed from $X$, and the $X$ boundary shrinks inward. We called that the internal p-set $X^{\overline{F}}$ is generated by $X$, $X^{\overline{F}} \subseteq X$. (b) If the attribute $\alpha_i$ is deleted from $\alpha$, $\alpha$ generates $\alpha^{\overline{F}}$, $\alpha^{\overline{F}} \subseteq \alpha$, then $X$ is supplemented with some elements, and the $X$ boundary is expanded outward. We call that $X$ generates the outer p-set $X^F$, $X \subseteq X^F$. (c) If you add some attributes into $\alpha$ and delete some other attributes from $\alpha$ at the same time, some elements are deleted from $X$ and some other elements are added into $X$. We call that $X$ generates a set pair $(X^{\overline{F}}, X^F)$ which is named by p-set. (d) If the above process continues, $X$ will generate multiple set pair $(X_1^{\overline{F}}, X_1^F), (X_2^{\overline{F}}, X_2^F), \cdots, (X_n^{\overline{F}}, X_n^F)$. We get the dynamic boundary of p-set: $X_n^{\overline{F}} \subseteq X_{n-1}^{\overline{F}} \subseteq \cdots \subseteq X_2^{\overline{F}} \subseteq X_1^{\overline{F}}, X_1^F \subseteq X_2^F \subseteq \cdots \subseteq X_n^F$. In the p-set, the attribute $\alpha_i$ of the element $x_i$ satisfies the expansion or contraction of "conjunctive normal form" in mathematical logic. For given the information $(x)$ which is defined by $X$, inner p-information $(x)^{\overline{F}}$, outer p-information $(x)^F$ and p-information $((x)^{\overline{F}}, (x)^F)$ are defined by $X^{\overline{F}}$, $X^F$ and $(X^{\overline{F}}, X^F)$ respectively, i.e., $(x) = X$, $(x)^{\overline{F}} = X^{\overline{F}}$, $(x)^F = X^F$, $((x)^{\overline{F}}, (x)^F) = (X^{\overline{F}}, X^F)$. We can speculate that p-sets can be used to analyze dynamic information recognition and information fusion. In fact, p-sets are the new mathematical methods and models for researching dynamic information recognition and fusion, because each information $(x)$ has an attribute set $\alpha$, that is, the information $(x)$ is associated with its attribute set $\alpha$. Given the existing researches that the p-set and p-augmented matrix have many applications in China [16–38] and some applications of function p-sets, the inverse p-sets and the inverse p-sets have made by many researchers [39–41].

In the actual data set, redundant information will inevitably appear. For example, the data collected by the sensor at a higher frequency is redundant for data analysis with a longer time span. Similarly, in information fusion, sometimes we need to add some information to improve the accuracy of the analysis. Therefore, we need to pay attention to redundant information fusion and supplementary information fusion. These two kinds of information fusion are more important in the era of big data. In this paper, two kinds of information fusion algorithms are proposed by analyzing p-augmented matrix reasoning from the dynamic boundary of p-set. The purpose of this paper is to improve the dynamic boundary of the p-set and its generated p-augmented matrix for information fusion based on the function p-sets, the inverse p-sets, and the function inverse p-sets. Compared with other traditional methods, p-set theory and method start from the attributes of data, through set operation, matrix reasoning, etc., obtain information equivalent classes, and mine unknown information.

The researches given in this paper are as follows: (a) we give the existing fact of the structure and logical features of p-sets, then we give the structure and generation method of p-augmented matrix. These concepts are preparations for reading this paper. (b) We analyze the dynamic boundary features and the generation of information equivalence classes of p-sets. (c) We give matrix reasoning intelligent acquisition and intelligent acquisition algorithm of information equivalence class generated by p-augmented matrix. (d) We analyze the relationships between the concepts of information equivalence class and information fusion. We find that information equivalence class and information fusion are equivalent. (e) We give the application of intelligent acquisition of information equivalence class on information fusion, which can be used in unknown information discovery.

## 2. Preparatory Concepts

Some preparatory concepts are given in literature [13–41].

### 2.1. The Structure of P-Sets and Their Logical Characteristics

Given a finite set of ordinary elements $X = \{x_1, x_2, \cdots, x_q\} \subset U$, $\alpha = \{\alpha_1, \alpha_2, \cdots, \alpha_k\} \subset V$ is a attribute set of $X$. $X^{\overline{F}}$ is called the internal p-set generated by $X$,

$$X^{\overline{F}} = X - X^-, \tag{1}$$

where $X^-$ is called the $\overline{F}$-deleted set of $X$,

$$X^- = \left\{ x_i | x_i \in X, \overline{f}(x_i) = u_i \overline{\in} X, \overline{f} \in \overline{F} \right\}. \tag{2}$$

If the attribute set $\alpha^F$ of $X^{\overline{F}}$ satisfies

$$\alpha^F = \alpha \cup \left\{ \alpha_i' | f(\beta_i) = \alpha_i' \in \alpha, f \in F \right\}, \tag{3}$$

where in (3), $\beta_i \in V$, $\beta_i \overline{\in} \alpha$, $f \in F$ turns $\beta_i$ into $f(\beta_i) = \alpha_i' \in \alpha$; in (1), $X^{\overline{F}} \neq \phi$, $X^{\overline{F}} = \{x_1, x_2, \cdots, x_p\}$, $p < q, p, q \in N^+$.

Given a finite set of ordinary elements $X = \{x_1, x_2, \cdots, x_q\} \subset U$, $\alpha = \{\alpha_1, \alpha_2, \cdots \alpha_k\} \subset V$ is the attribute set of $X$. $X^F$ is called outer p-set generated by $X$,

$$X^F = X \cup X^+, \tag{4}$$

where $X^+$ is called $F$-supplemented set of $X$,

$$X^+ = \left\{ u_i | u_i \in U, u_i \overline{\in} X, f(u_i) = x_i' \in X, f \in F \right\}. \tag{5}$$

If the attribute set $\alpha^{\overline{F}}$ of $X^F$ satisfies

$$\alpha^{\overline{F}} = \alpha - \left\{ \beta_i | \overline{f}(\alpha_i) = \beta_i \overline{\in} \alpha, \overline{f} \in \overline{F} \right\}, \tag{6}$$

where in (6), $\alpha_i \in \alpha$, $\overline{f} \in \overline{F}$ turns $\alpha_i$ into $\overline{f}(\alpha_i)$, $\overline{f}(\alpha_i) = \beta_i \overline{\in} \alpha$; in (6), $\alpha^{\overline{F}} \neq \phi$; in (4), $X^F = \{x_1, x_2, \cdots, x_r\}$, $q < r, q, r \in N^+$.

The finite ordinary element set pair composed by internal p-set $X^{\overline{F}}$ and outer p-set $X^F$ is called p-set generated by $X$, namely

$$(X^{\overline{F}}, X^F). \tag{7}$$

The finite ordinary element set $X$ is called the base set of p-set $(X^{\overline{F}}, X^F)$.

It is obtained from (3) that

$$\alpha_1^F \subseteq \alpha_2^F \subseteq \cdots \subseteq \alpha_{n-1}^F \subseteq \alpha_n^F. \tag{8}$$

Internal p-sets can be obtained accordingly from (1), (8) as following:

$$X_n^{\overline{F}} \subseteq X_{n-1}^{\overline{F}} \subseteq \cdots \subseteq X_2^{\overline{F}} \subseteq X_1^{\overline{F}}. \tag{9}$$

It is obtained from (6) that

$$\alpha_n^{\overline{F}} \subseteq \alpha_{n-1}^{\overline{F}} \subseteq \cdots \subseteq \alpha_2^{\overline{F}} \subseteq \alpha_1^{\overline{F}}. \tag{10}$$

Outer p-sets can be obtained accordingly from (4) and (10) as follows:

$$X_1^F \subseteq X_2^F \subseteq \cdots \subseteq X_{n-1}^F \subseteq X_n^F. \tag{11}$$

By using (9) and (11), the set is obtained as follow:

$$\left\{ (X_i^{\overline{F}}, X_j^F) | i \in I, j \in J \right\}, \tag{12}$$

which is called the p-set family generated by $X$, and (12) is the general form of the p-set.

Some theorems can be obtained from (1)–(7), (12) as following:

**Theorem 1.** *If $F = \overline{F} = \phi$, then the p-set $(X^{\overline{F}}, X^F)$ is restored to the finite ordinary element set X, namely*

$$(X^{\overline{F}}, X^F)_{F=\overline{F}=\phi} = X. \tag{13}$$

**Theorem 2.** *If $F = \overline{F} = \phi$, then the p-set family $\{(X_i^{\overline{F}}, X_j^F)|i \in I, j \in J\}$ is restored to a finite set of ordinary element set X, namely*

$$\left\{(X_i^{\overline{F}}, X_j^F)|i \in I, j \in J\right\}_{F=\overline{F}=\phi} = X. \tag{14}$$

Special notes:

1.  $U$ is the finite element universe, and $V$ is the finite attribute universe.
2.  $F = \{f_1, f_2, \cdots, f_n\}$, $\overline{F} = \left\{\overline{f}_1, \overline{f}_2, \cdots, \overline{f}_n\right\}$ are element or attribute transfer families; $f \in F$, $\overline{f} \in \overline{F}$ are element or attribute transfer; element (or attribute) transfer is a function concept of transformation.
3.  The characteristic of $f \in F$ is that, for the element $u_i \in U$, $u_i \overline{\in} X$, $f \in F$ turns $u_i$ into $f(u_i) = x_i' \in X$; for the attribute $\beta_i \in V$, $\beta_i \overline{\in} \alpha$, $f \in F$ turns $\beta_i$ into $f(\beta_i) = \alpha_i' \in \alpha$.
4.  The characteristic of $\overline{f} \in \overline{F}$ is that: for element $x_i \in X$, $\overline{f} \in \overline{F}$ turns $x_i$ into $\overline{f}(x_i) = u_i \overline{\in} X$; for the attribute $\alpha_i \in \alpha$, $\overline{f} \in \overline{F}$ turns $\alpha_i$ into $\overline{f}(\alpha_i) = \beta_i \overline{\in} \alpha$.
5.  The dynamic feature of the Equation (1) is the same as the dynamic feature of the inverse accumulator $T = T - 1$.
6.  The dynamic feature of Equation (4) is the same as the dynamic feature of the accumulator $T = T + 1$. For example, in Equation (4), $X_1^F = X \cup X_1^+$, let $X = X_1^F$, then $X_2^F = X_1^F \cup X_2^+ = (X \cup X_1^+) \cup X_2^+, \cdots$, and so on.

*2.2. The Existence Fact of P-Sets and Its Logical Characteristics*

Suppose that $X = \{x_1, x_2, x_3, x_4, x_5\}$ is a set of finite ordinary elements in which there are 5 apples, $\alpha = \{\alpha_1, \alpha_2, \alpha_3\}$ is a attribute set of X, $\alpha_1$=Red, $\alpha_2$=Sweet, $\alpha_3$=Red Fuji; $\forall x_i \in X$, $x_i$ have the attributes $\alpha_1$, $\alpha_2$ and $\alpha_3$. By using the "conjunctive normal form" in mathematical logic, we can obtain the following facts:

Given the attribute $\alpha_i$ for $\forall x_i \in X$, $\alpha_i = \alpha_1 \wedge \alpha_2 \wedge \alpha_3, i = 1, 2, 3, 4, 5$,

1.  If $\alpha_4 = produced\ from\ Yantai,\ Chinese$ is added to $\alpha$, $\alpha$ generates $\alpha^F$, $\alpha \subseteq \alpha^F$, $\alpha^F = \alpha \cup \{\alpha_4\} = \{\alpha_1, \alpha_2, \alpha_3, \alpha_4\}$, then $x_4$, $x_5$ are deleted from X, X generates internal p-set $X^{\overline{F}}$, $X^{\overline{F}} \subseteq X$, $X^{\overline{F}} = X - \{x_4, x_5\} = \{x_1, x_2, x_3\}$, the attribute $\alpha_i$ for $\forall x_i \in X^{\overline{F}}$ satisfies $\alpha_i = (\alpha_1 \wedge \alpha_2 \wedge \alpha_3) \wedge \alpha_4 = \alpha_1 \wedge \alpha_2 \wedge \alpha_3 \wedge \alpha_4; i = 1, 2, 3$.
2.  If the attribute $\alpha_3$ is deleted in $\alpha$, $\alpha$ generates $\alpha^{\overline{F}}$, $\alpha^{\overline{F}} \subseteq \alpha$, $\alpha^{\overline{F}} = \alpha - \{\alpha_3\} = \{\alpha_1, \alpha_2\}$, then $x_6$, $x_7$ is supplemented to X, X generates an outer p-set $X^F$, $X \subseteq X^F$, $X^F = X \cup \{x_6, x_7\} = \{x_1, x_2, x_3, x_4, x_5, x_6, x_7\}$, the attribute $\alpha_i$ for $\forall x_i \in X^F$ satisfies $\alpha_i = (\alpha_1 \wedge \alpha_2 \wedge \alpha_3) - \wedge \alpha_3 = \alpha_1 \wedge \alpha_2, i = 1, 2, 3, 4, 5, 6, 7$.
3.  If you add some attributes into $\alpha$ and delete some other attributes from $\alpha$ at the same time, $\alpha$ generates $\alpha^F$ and $\alpha^{\overline{F}}$, i.e., $\alpha$ generates $(\alpha^F, \alpha^{\overline{F}})$, then X generates $X^{\overline{F}}$ and $X^F$, i.e., X generates a p-set $(X^{\overline{F}}, X^F)$.
4.  If the process of adding some attributes into $\alpha$ while deleting other attributes continues from $\alpha$, X generates multiple p-sets: $(X_1^{\overline{F}}, X_1^F), (X_2^{\overline{F}}, X_2^F), \cdots, (X_n^{\overline{F}}, X_n^F)$, which are the p-set family which is showed as Equation (12).

For $X = \{x_1, x_2, \cdots x_q\}$, $\alpha = \{\alpha_1, \alpha_2, \cdots, \alpha_\eta, \alpha_{\eta+1}, \cdots \alpha_k\}$ is the attribute set of X; for $X^{\overline{F}} = \{x_1, x_2, \cdots x_p\}$, $\alpha^F = \{\alpha_1, \alpha_2, \cdots, \alpha_k, \alpha_{k+1}, \cdots, \alpha_\lambda\}$ is the attribute set of $X^{\overline{F}}$; for $X^F = \{x_1, x_2, \cdots, x_r\}$, $\alpha^{\overline{F}} = \{\alpha_1, \alpha_2, \cdots, \alpha_\eta\}$ is the attribute set of $X^F$; $p < q < r$, $p, q, r \in N^+$; $\eta < k < \lambda$, $\eta, k, \Lambda \in N^+$. Some general conclusions can be obtained from the above facts 1–4 as following:

1. The attribute $\alpha_i$ for $\forall x_i \in X$ satisfies the attribute's conjunctive normal form:

$$\alpha_i = \wedge_{t=1}^{k}\alpha_t. \tag{15}$$

2. The attribute $\alpha_i$ for $\forall x_i \in X^{\overline{F}}$ satisfies the expansion of attribute's conjunctive normal form:

$$\alpha_i = \left(\wedge_{t=1}^{k}\alpha_t\right) \wedge_{t=k+1}^{\lambda} \alpha_t. \tag{16}$$

3. The attribute $\alpha_i$ for $\forall x_i \in X^{F}$ satisfies the contraction of attribute's conjunctive normal form:

$$\alpha_i = \left(\wedge_{t=1}^{k}\alpha_t\right) - \wedge_{t=\eta+1}^{k}\alpha_t. \tag{17}$$

4. The attribute $\alpha_i$ for $\forall x_i \in X^{\overline{F}}$ and the attribute $\alpha_j$ for $\forall x_j \in X^{F}$ satisfies the expansion and contraction of attribute's conjunctive normal form:

$$(\alpha_i, \alpha_j) = \left(\left(\wedge_{t=1}^{k}\alpha_t\right) \wedge_{t=k+1}^{\lambda} \alpha_t, \left(\wedge_{t=1}^{k}\alpha_t\right) - \wedge_{t=\eta+1}^{k}\alpha_t\right), \tag{18}$$

where, $\alpha_i = \left(\wedge_{t=1}^{k}\alpha_t\right)\wedge_{t=k+1}^{\lambda}\alpha_t$, $\alpha_j = \left(\wedge_{t=1}^{k}\alpha_t\right) - \wedge_{t=\eta+1}^{k}\alpha_t$; Equations (15)–(18) are the logical feature of the p-set $(X^{\overline{F}}, X^{F})$.

### 2.3. Structure and Generation of P-Augmented Matrix

By using the structure of the p-set, the definition and structure of improved general augmentation matrix $A^*$ are given in literature [38]:

Given a finite set of ordinary elements $X = \{x_1, x_2, \cdots, x_q\}$, $x_i$ ($\forall x_i \in X$) has $n$ values $y_{i,1}, y_{i,2}, \cdots, y_{i,n}$; $y_j = (y_{i,1}, y_{i,2}, \cdots, y_{i,n})^T$ is a vector generated by $y_{i,1}, y_{i,2}, \cdots, y_{i,n}$, the matrix $A$ can be obtained by using $y_i$ as the column. The $A$ is called element value matrix generated by $X$

$$A = \begin{bmatrix} Y_{1,1} & y_{1,2} & \cdots & y_{1,q} \\ Y_{2,1} & y_{2,2} & \cdots & y_{2,q} \\ \vdots & \vdots & \ddots & \vdots \\ Y_{n,1} & y_{n,2} & \cdots & y_{n,q} \end{bmatrix}. \tag{19}$$

The $A^{\overline{F}}$ is called the internal p-augmented matrix of $A$ generated by internal p-set $X^{\overline{F}} = \{x_1, x_2, \cdots, x_p\}$,

$$A^{\overline{F}} = \begin{bmatrix} Y_{1,1} & y_{1,2} & \cdots & y_{1,p} \\ Y_{2,1} & y_{2,2} & \cdots & y_{2,p} \\ \vdots & \vdots & \ddots & \vdots \\ Y_{n,1} & y_{n,2} & \cdots & y_{n,p} \end{bmatrix}. \tag{20}$$

The $A^{F}$ is called the outer p-augmented matrix of $A$ generated by the outer p-set $X^{F} = \{x_1, x_2, \cdots, x_r\}$,

$$A^{F} = \begin{bmatrix} Y_{1,1} & y_{1,2} & \cdots & y_{1,r} \\ Y_{2,1} & y_{2,2} & \cdots & y_{2,r} \\ \vdots & \vdots & \ddots & \vdots \\ Y_{n,1} & y_{n,2} & \cdots & y_{n,r} \end{bmatrix}. \tag{21}$$

The matrix pair consisting of the inner p-augmented matrix $A^{\overline{F}}$ and outer p-augmented matrix $A^{F}$ is as following

$$(A^{\overline{F}}, A^{F}). \tag{22}$$

The $(A^{\overline{F}}, A^F)$ is called p-augmented matrix of $A$ generated by p-set $(X^{\overline{F}}, X^F)$, where, in Equations (19)–(21), $p < q < r$, $p, q, r \in N^+$. The outer p-augmented matrix $A^F$ of $A$ is the same concept as the ordinary augmentation matrix $A^*$ of $A$.

Figure 1 shows a two-dimensional visual representation of the p-set $(X^{\overline{F}}, X^F)$.

The following conclusions are directly obtained from Equations (1)–(7) and the Figure 1:

1. The $X$ boundary is contracting inward when some attributes are added into the attribute set $\alpha$ of $X$. That is, the $X$ dynamically generates the internal p-set $X^{\overline{F}}$.
2. The $X$ boundary is expanding outward when some attributes are deleted from the attribute set $\alpha$ of $X$. That is, the $X$ dynamically generates the outer p-set $X^F$.
3. The boundary of $X$ is contracting inward and expanding outward when some attributes are added and some attributes are deleted in attribute collection $\alpha$ of $X$. That is, the $X$ dynamically generates p-set $(X^{\overline{F}}, X^F)$; the process of adding attributes and deleting attributes in $\alpha$ keeps going, $X$ dynamically generates p-set families.

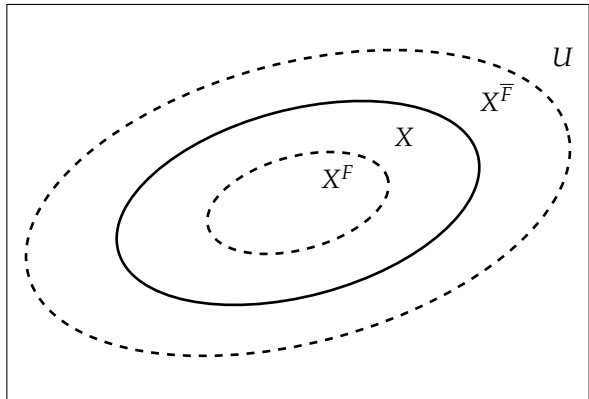

**Figure 1.** The positional relationship between the finite ordinary element set $X$ and the p-set $(X^{\overline{F}}, X^F)$, $X^{\overline{F}} \subseteq X \subseteq X^F$; where, $X$ is represented by a solid line, $X^{\overline{F}}$ and $X^F$ are indicated by dashed lines respectively; the p-set is composed of $X^{\overline{F}}$ and $X^F$.

The concepts in this section are important for accepting the research and results given in Sections 3–5. More features and applications of p-sets and p-augmented matrices can be found from the works of literature [13–38].

Convention: $X$, $X^{\overline{F}}$, $X^F$ and $(X^{\overline{F}}, X^F$ are defined as the information $(x)$, the inner p-information $(x)^{\overline{F}}$, the outer p-information $(x)^F$ and the p-information $((x)^{\overline{F}}, X^F)$ respectively; i.e., $(x) = X$, $(x)^{\overline{F}} = X^{\overline{F}}$, $(x)^F = X^F$ and $((x)^{\overline{F}}, X^F) = (X^{\overline{F}}, X^F)$. These concepts and symbols are used in Sections 3–6.

## 3. Dynamic Boundary of P-Sets and Dynamic Generation of Information Equivalence Classes

**Theorem 3.** *(The dynamic generation theorem of $\alpha^F$-information equivalence class $[x]^{\overline{F}}$) if some attributes are added into the attribute set $\alpha$ of information $(x)$, $\alpha$ generates $\alpha^F$, $\alpha \subseteq \alpha^F$, then the internal p-information $(x)^{\overline{F}}$ with the attribute set $\alpha^F$ is the $\alpha^F$-information equivalence class $[x]^{\overline{F}}$ generated by $(x)$. That is*

$$(x)^{\overline{F}} = [x]^{\overline{F}}. \tag{23}$$

**Proof.** Suppose that $(x)^{\overline{F}}$ is the internal p-information generated by the information $(x)$, the attribute set $\alpha^F$ of $(x)^{\overline{F}}$ is the relationship $R$ of $(x)^{\overline{F}} \times (x)^{\overline{F}}$, i.e., $R = \alpha^F$; Some equivalence class concepts can be obtained: 1. For $\forall x_i \in (x)^{\overline{F}}$, $x_i$ and $x_i$ have the relationship $R$, i.e., $x_i \alpha^F x_j$, so the reflexivity is satisfied. 2. For $\forall x_i, x_j \in (x)^{\overline{F}}$, $x_i$ has a relationship $R$ with $x_j$, then $X_j$ has a relationship $R$ with $x_i$; i.e., if $x_i \alpha^F x_j$, then $x_j \alpha^F x_i$ could be obtained, so the symmetry is satisfied. 3. For $\forall x_i, x_j, x_k \in (x)^{\overline{F}}$, if $x_i$ has a relationship $R$ with $x_j$, and $x_j$ has a relationship $R$ with $x_k$, then $x_i$ has a relationship $R$

with $x_k$; i.e., if $x_i\alpha^F x_j$, and $x_j\alpha^F x_k$, then the $x_i\alpha^F x_k$ could be obtained. So the transitivity is satisfied. From 1–3 we can obtained that: for $\forall x_i, x_j, x_k \in (x)^{\overline{F}}$, $\alpha^F$ satisfies with the reflexivity $x_i\alpha^F x_i$; the symmetry $x_i\alpha^F x_j \Rightarrow x_j\alpha^F x_i$; and the transitivity $x_i\alpha^F x_j, x_j\alpha^F x_k \Rightarrow x_i\alpha^F x_k$. It is easy to get that: internal p-information $(x)^{\overline{F}}$ is the $\alpha^F$- information equivalence class $[x]^{\overline{F}}$ generated by the information $(x)$, $(x)^{\overline{F}} = [x]^{\overline{F}}$.   □

**Theorem 4.** *(The dynamic generation theorem of $\alpha^{\overline{F}}$-Information equivalence class $[x]^F$ ) If some attributes are deleted from attribute set $\alpha$ of information $(x)$, $\alpha$ generates $\alpha^{\overline{F}}$, $\alpha^F \subseteq \alpha$, then the outer p-information $(x)^F$ with attribute set $\alpha^{\overline{F}}$ is the $\alpha^F$- Information equivalence class $[x]^F$ generated by $(x)$; that is,*

$$(x)^F = [x]^F. \tag{24}$$

The proof is similar to Theorem 1, so the proof of Theorem 2 is omitted.
From Theorems 3 and 4, the Theorem 5 can be obtained directly,

**Theorem 5.** *(The dynamic generation theorem of $(\alpha^{\overline{F}},\alpha^F)$- information equivalence class $[[x]^{\overline{F}}, [x]^F]$ ) If some attributes are added to and deleted from attribute set $\alpha$ of information $(x)$ at the same time, $\alpha$ generates $\alpha^F$ and $\alpha^{\overline{F}}$, $\alpha^{\overline{F}} \subseteq \alpha \subseteq \alpha^F$, then the p-information $((x)^{\overline{F}}, (x)^F)$ with attribute set $(\alpha^F, \alpha^{\overline{F}})$ is the $(\alpha^F, \alpha^{\overline{F}})$-Information equivalence class $[[x]^{\overline{F}}, [x]^F]$ generated by$(x)$; that is,*

$$((x)^{\overline{F}}, (x)^F) = [[x]^{\overline{F}}, [x]^F]. \tag{25}$$

Obviously, the information $(x)$ with the attribute set $\alpha$ is the $\alpha$-information equivalence class $[x]$, $[x] = (x)$.

Some propositions can be obtained from Theorems 3–5 and Equations (1)–(7) in Section 2 as following:

**Proposition 1.** *The dynamic generation of $\alpha^F$- information equivalence class $[x]^{\overline{F}}$ is synchronous with the boundary inward dynamic contraction of the internal p-set $X^{\overline{F}}$.*

**Proposition 2.** *The dynamically generation of $\alpha^{\overline{F}}$- information equivalence class $[x]^F$ is synchronous with the boundary outward expansion of the outer p-set $X^F$.*

**Proposition 3.** *The dynamically generation of $(\alpha^F, \alpha^{\overline{F}})$-Information equivalence class $[[x]^{\overline{F}}, [x]^F]$ are synchronous with the boundary inward dynamic contraction and outward dynamic expansion of the p-set $(X^{\overline{F}}, X^F)$.*

## 4. Matrix Reasoning and the Intelligent Acquisition Theorem of Information Equivalence Classes

Conventions: in Section 2, the internal p-augmented matrix $A^{\overline{F}}$, outer p-augmented matrix $A^F$ and p-augmented matrix $(A^{\overline{F}}, A^F)$ are recorded as the internal p-matrix $A^{\overline{F}}$, the outer p-matrix $A^F$ and p-matrix $(A^{\overline{F}}, A^F)$ respectively. It will not cause any misunderstanding.

Given internal p-matrix $A_k^{\overline{F}}$ and $A_{k+1}^{\overline{F}}$, $\alpha_k^F$, $\alpha_{k+1}^F$ are the attribute set of $A_k^{\overline{F}}$, $A_{k+1}^{\overline{F}}$ respectively; $A_k^{\overline{F}}$, $A_{k+1}^{\overline{F}}$ and $\alpha_k^F$, $\alpha_{k+1}^F$ satisfy the following equation

$$if \quad A_{k+1}^{\overline{F}} \Rightarrow A_k^{\overline{F}}, \quad then \quad \alpha_k^F \Rightarrow \alpha_{k+1}^F. \tag{26}$$

Equation (26) is called internal p-matrix reasoning generated by internal p-matrix; $A_{k+1}^{\overline{F}} \Rightarrow A_k^{\overline{F}}$ is called the internal p-matrix reasoning condition, $\alpha_k^F \Rightarrow \alpha_{k+1}^F$ is called the internal p-matrix reasoning conclusion. Where, in Equation (26), $A_{k+1}^{\overline{F}} \Rightarrow A_k^{\overline{F}}$ is equivalent to $A_{k+1}^{\overline{F}} \subseteq A_k^{\overline{F}}$; $\alpha_k^F \Rightarrow \alpha_{k+1}^F$ is equivalent to $\alpha_k^F \subseteq \alpha_{k+1}^F$.

Given the outer p-matrix $A_k^F$ and $A_{k+1}^F$, $\alpha_k^{\overline{F}}$ and $\alpha_{k+1}^{\overline{F}}$ are the attribute set of $A_k^F$, $A_{k+1}^F$, respectively. $A_k^F$, $A_{k+1}^F$ and $\alpha_k^{\overline{F}}$, $\alpha_{k+1}^{\overline{F}}$ satisfy the following equation

$$if \quad A_k^F \Rightarrow A_{k+1}^F, \quad then \quad \alpha_{k+1}^{\overline{F}} \Rightarrow \alpha_k^{\overline{F}}. \tag{27}$$

Equation (27) is called outer p-matrix reasoning generated by outer p-matrix; $A_k^F \Rightarrow A_{k+1}^F$ is called outer p-matrix inference condition, $\alpha_{k+1}^{\overline{F}} \Rightarrow \alpha_k^{\overline{F}}$ is called the outer p-matrix reasoning conclusion.

Given p-matrix $(A_{k+1}^{\overline{F}}, A_k^F)$ and $(A_k^{\overline{F}}, A_{k+1}^F)$, $(\alpha_{k+1}^F, \alpha_k^{\overline{F}})$ and $(\alpha_k^F, \alpha_{k+1}^{\overline{F}})$ are the attribute sets of $(A_{k+1}^{\overline{F}}, A_k^F)$, $(A_k^{\overline{F}}, A_{k+1}^F)$ respectively. $(A_{k+1}^{\overline{F}}, A_k^F)$, $(A_k^{\overline{F}}, A_{k+1}^F)$, $(\alpha_{k+1}^F, \alpha_k^{\overline{F}})$ and $(\alpha_k^F, \alpha_{k+1}^{\overline{F}})$ satisfy the following equation

$$
\begin{aligned}
if \quad & (A_{k+1}^{\overline{F}}, A_k^F) \Rightarrow (A_k^{\overline{F}}, A_{k+1}^F), \\
then \quad & (\alpha_k^F, \alpha_{k+1}^{\overline{F}}) \Rightarrow (\alpha_{k+1}^F, \alpha_k^{\overline{F}}).
\end{aligned} \tag{28}
$$

Equation (28) is called p-matrix reasoning generated by p-matrix; $(A_{k+1}^{\overline{F}}, A_k^F) \Rightarrow (A_k^{\overline{F}}, A_{k+1}^F)$ is called the p-matrix reasoning condition, $(\alpha_k^F, \alpha_{k+1}^{\overline{F}}) \Rightarrow (\alpha_{k+1}^F, \alpha_k^{\overline{F}})$ is called p-matrix reasoning conclusion. Where, in Equation (28), $(A_{k+1}^{\overline{F}}, A_k^F) \Rightarrow (A_k^{\overline{F}}, A_{k+1}^F)$ means that $A_{K+1}^{\overline{F}} \Rightarrow A_K^{\overline{F}}$, $A_k^F \Rightarrow A_{k+1}^F$.

There are some special explanation: from Section 2, $A_{k+1}^{\overline{F}}$ is generated by the value of $X_{k+1}^{\overline{F}}$; $A_{k+1}^{\overline{F}}$ does not change the attribute set of $X_{k+1}^{\overline{F}}$; $A_{k+1}^{\overline{F}}$ and $X_{k+1}^{\overline{F}}$ have the same attribute set $\alpha_{k+1}^{\overline{F}}$; $A_k^F$ is generated by the value of $X_k^F$; $A_k^F$ does not change the attribute set of $X_k^F$; $A_k^F$ and $X_k^F$ have the same attribute set $\alpha_k^{\overline{F}}$.

From Equations (26)–(28), we can obtain

**Theorem 6.** *(The intelligent acquisition theorem of $\alpha^F$-Information equivalence class $[x]^{\overline{F}}$) if the internal p-matrix $A_k^{\overline{F}}$, $A_{k+1}^{\overline{F}}$ and $\alpha^F$-Information equivalence class $[x]_k^{\overline{F}}$, $[x]_{k+1}^{\overline{F}}$ satisfy*

$$if \quad A_{k+1}^{\overline{F}} \Rightarrow A_k^{\overline{F}}, \quad then \quad [x]_{k+1}^{\overline{F}} \Rightarrow [x]_k^{\overline{F}}. \tag{29}$$

*Then, under the condition of $A_{k+1}^{\overline{F}} \Rightarrow A_k^{\overline{F}}$, $\alpha^F$-information equivalence class $[x]_{k+1}^{\overline{F}}$ is acquired intelligently from $[x]_k^{\overline{F}}$; $[x]_{k+1}^{\overline{F}} \subseteq [x]_k^{\overline{F}}$.*

**Proof.** From Section 2, we obtained that: $A_{k+1}^{\overline{F}}$, $A_k^{\overline{F}}$ are generated by $(x)_{k+1}^{\overline{F}}$, $(x)_k^{\overline{F}}$ respectively; $A_{k+1}^{\overline{F}}$ and $A_k^{\overline{F}}$ satisfy $A_{k+1}^{\overline{F}} \subseteq A_k^{\overline{F}}$, that is, $A_{k+1}^{\overline{F}} \Rightarrow A_k^{\overline{F}}$. By using Theorem 3, we get that: $[x]_{k+1}^{\overline{F}}$, $[x]_k^{\overline{F}}$ are the $\alpha^F$-information equivalent equivalence class generated by information $(x)$. $[x]_{k+1}^{\overline{F}}$ and $[x]_k^{\overline{F}}$ satisfy $[x]_{k+1}^{\overline{F}} \subseteq [x]_k^{\overline{F}}$, that is, $[x]_{k+1}^{\overline{F}} \Rightarrow [x]_k^{\overline{F}}$. Under the internal p-matrix reasoning condition $A_{k+1}^{\overline{F}} \Rightarrow A_k^{\overline{F}}$, $[x]_{k+1}^{\overline{F}} \Rightarrow [x]_k^{\overline{F}}$ is obtained, that is, $[x]_{k+1}^{\overline{F}} \subseteq [x]_k^{\overline{F}}$. $[x]_{k+1}^{\overline{F}}$ is acquired intelligently in $[x]_k^{\overline{F}}$. □

**Theorem 7.** *(The intelligent acquisition theorem of $\alpha^{\overline{F}}$-Information equivalence class $[x]^F$) If the outer p-matrix $A_k^F$, $A_{k+1}^F$ and $\alpha^{\overline{F}}$-Information equivalence class $[x]_k^F$, $[x]_{k+1}^F$ satisfy*

$$if \ A_k^F \Rightarrow A_{k+1}^F, \ then \ [x]_k^F \Rightarrow [x]_{k+1}^F. \tag{30}$$

*Then, under the condition of $A_k^F \Rightarrow A_{k+1}^F$, $\alpha^{\overline{F}}$-information equivalence class $[x]_{k+1}^F$ is acquired intelligently by $[x]_k^F$; $[x]_k^F \subseteq [x]_{k+1}^F$.*

The proof of Theorem 7 is similar to Theorem 6, so the proof is omitted.

From Theorems 6 and 7, we can obtained directly the following theorem:

**Theorem 8.** *(The intelligent acquisition theorem of $(\alpha^F, \alpha^{\overline{F}})$-information equivalence class $([x]^{\overline{F}}, [x]^F)$)*
*If the p-matrix $(A^{\overline{F}}_{k+1}, A^F_k)$, $(A^{\overline{F}}_k, A^F_{k+1})$ and $(\alpha^{\overline{F}}, \alpha^F)$-information equivalence class $[[x]^{\overline{F}}_{k+1}, [x]^F_k]$, $[[x]^{\overline{F}}_k, [x]^F_{k+1}]$ satisfy*

$$if \quad (A^{\overline{F}}_{k+1}, A^F_k) \Rightarrow (A^{\overline{F}}_k, A^F_{k+1}),$$
$$then \quad [[x]^{\overline{F}}_{k+1}, [x]^F_k] \Rightarrow [[x]^{\overline{F}}_k, [x]^F_{k+1}].$$

(31)

*Then under the condition of $(A^{\overline{F}}_{k+1}, A^F_k) \Rightarrow (A^{\overline{F}}_k, A^F_{k+1})$, the $[x]^{\overline{F}}_{k+1}, [x]^F_{k+1}$ in the $(\alpha^F, \alpha^{\overline{F}})$-information equivalence class $[[x]^{\overline{F}}_{k+1}, [x]^F_{k+1}]$ will be acquired intelligently in $[x]^{\overline{F}}_k$ and $[x]^F_k$ respectively. $[x]^{\overline{F}}_{k+1} \subseteq [x]^{\overline{F}}_k$, $[x]^F_k \subseteq [x]^F_{k+1}$.*

**Corollary 1.** *If $[x]^{\overline{F}}$ is the $\alpha^F$-information equivalence class generated intelligently by the internal p-matrix reasoning, $[x]^{\overline{F}} \subseteq (x)$, then the attribute set $\alpha$ of the $(x)$ must be supplemented with some attributes $\alpha_i$.*

The proof is obtained directly by Theorem 3, and the proof of Corollary 1 is omitted.

**Corollary 2.** *If $[x]^F$ is the $\alpha^{\overline{F}}$- information equivalence class generated intelligently by outer p-matrix reasoning, $(x) \subseteq [x]^F$, then the attribute set $\alpha$ of information $(x)$ must be deleted some attributes $\alpha_j$.*

The proof of Corollary 2 is similar to Corollary 1, and the proof is omitted.
From Corollaries 1 and 2, we can obtain directly the following corollary:

**Corollary 3.** *If $[[x]^{\overline{F}}, [x]^F]$ is the $(\alpha^F, \alpha^{\overline{F}})$-information equivalence class generated intelligently by p-matrix reasoning, $[x]^{\overline{F}} \subseteq (x)$, $(x) \subseteq [x]^F$, then the attribute set $\alpha$ of information $(x)$ must be added into the attributes $\alpha_i$ and must be delete the attribute $\alpha_j$.*

The intelligent acquisition algorithm of information acquisition class can be obtained by using the concepts and results given in Section 4 (showed in Figure 2). It should be noted that the intelligent algorithm diagram of $\alpha^{\overline{F}}$- information equivalence class $[x]^F_k$ is similar to Figure 2. It is omitted.

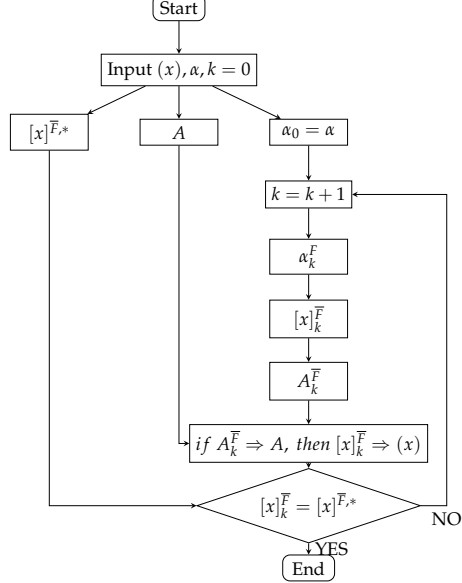

**Figure 2.** The intelligent algorithm diagram of $\alpha^F$-information equivalence class $[x]^{\overline{F}}$. In the figure, $(x)$ is the given information; $\alpha$ is the attribute set of $(x)$; $[x]^{\overline{F}}_k$ is $\alpha^F_k$-information equivalence class; $[x]^{\overline{F},*}$ is the given $\alpha^F$- information equivalence class; $A^{\overline{F}}_k$, $A^{\overline{F}}_{k+t}$ are the internal p-matrix generated by $[x]^{\overline{F}}_k$, $[x]^{\overline{F}}_{k+t}$ respectively; $A$ is the information value matrix generated by $(x)$.

## 5. The Relationship between Information Equivalence and Information Fusion

### 5.1. Two Types of Information Fusion

For example, there are two boxes A and B on the table; there are $m$ grains of soybeans in box A, and $n$ grains of wheat in box B.

I. Children $w$ puts $n$ grains of wheat in box B into box A, then the $m$ grains of soybeans mixed with $n$ grains of wheat.

II. Children $w$ pour the mixture of $m$ grains and $n$ grains into a sieve. The wheat grains are filtered by a sieve and separated from it, and $m$ grains of soybean are left in the sieve.

If $m$ grains of soybeans in box A are considered as $m$ information elements $x_i$, $n$ grains of wheat in box B are considered as $n$ information elements $x_j$, $x_i \neq x_j$. The two conclusions I* and II* are obtained by using the concept of information fusion to understand the above facts I and II as following:

I*. The $n$ information elements $x_j$ are merged into A from outside A, which generates information fusion $(x)^F$. There are $m + n$ information elements $x_k$ in $(x)^F$; $(x)^F$ is the first type of information fusion. The first type of information fusion is called information supplementation fusion.

II*. The $n$ information elements $x_j$ among the $m + n$ information elements in box A are transfer from inside to outside of box A, which generates information fusion $(x)^{\overline{F}}$. There are $m$ information elements $x_i$ in $(x)^{\overline{F}}$. $(x)^{\overline{F}}$ is the second type of information fusion. The second type of information fusion is called information redundancy fusion. The characteristics of the two types of information fusion are exactly the same as those of the p-set $(X^{\overline{F}}, X^F)$. P-set is a new model and new method for researching information fusion.

### 5.2. The Relationship between Two Types of Information Fusion and Information Equivalence Classes

From the above simple example, we analyze the relationship among two types of information fusion and information equivalence class.

The information supplementation fusion $(x)^F$ is called $\alpha^{\overline{F}}$-information equivalence class $[x]^F$ on the attribute set $\alpha^{\overline{F}}$, if $(x)^F$ is the generation of the delete attribute in the attribute set $\alpha$ of information $(x)$.

The information redundancy fusion $(x)^{\overline{F}}$ is called $\alpha^F$-information equivalence class $[x]^{\overline{F}}$ on the attribute set $\alpha^F$, if $(x)^{\overline{F}}$ is the generation of the supplementary attribute in the attribute set $\alpha$ of information $(x)$.

The information fusion pair $((x)^{\overline{F}}, (x)^F)$ is composed of information redundancy fusion $(x)^{\overline{F}}$ and information supplementation fusion $(x)^F$. $((x)^{\overline{F}}, (x)^F)$ is called the $(\alpha^F, \alpha^{\overline{F}})$-information equivalence class $[[x]^{\overline{F}}, [x]^F]$ on the attribute set $(\alpha^F, \alpha^{\overline{F}})$.

By using these concepts, we can get the following theorems.

**Theorem 9.** *(The relationship theorem of information supplementation fusion and $\alpha^{\overline{F}}$- information equivalence class) Information supplementation fusion $(x)^F$ is the $\alpha^{\overline{F}}$- information equivalence class generated by information $(x)$ if and only if $\forall\, x_i, x_j, X_k \in (x)^F$ satisfy*

$$
\begin{aligned}
&1.\ \textit{reflexivity.}\quad x_i \alpha^{\overline{F}} x_i \\
&2.\ \textit{symmetry.}\quad x_i \alpha^{\overline{F}} x_j \Rightarrow x_j \alpha^{\overline{F}} x_i \\
&3.\ \textit{transitivity.}\quad x_i \alpha^{\overline{F}} x_j, x_j \alpha^{\overline{F}} x_k \Rightarrow x_i \alpha^{\overline{F}} x_k.
\end{aligned}
\tag{32}
$$

**Theorem 10.** *(The relationship theorem of information redundancy fusion and $\alpha^F$-information equivalence class) Information redundancy fusion $(x)^{\overline{F}}$ is the $\alpha^F$-information equivalence class $[x]^{\overline{F}}$ generated by information $(x)$ if and only if $\forall x_i, x_j, x_k \in (x)^{\overline{F}}$ satisfy*

$$
\begin{aligned}
&1.\ \textit{reflexivity.} && x_i \alpha^F x_i \\
&2.\ \textit{symmetry.} && x_i \alpha^F x_j \Rightarrow x_j \alpha^F x_i \\
&3.\ \textit{transitivity.} && x_i \alpha^F x_j, x_j \alpha^F x_k \Rightarrow x_i \alpha^F x_k.
\end{aligned}
\tag{33}
$$

Theorems 9 and 10 can be obtained directly by using Theorem 3. The proof of Theorems 9 and 10 is omitted.

**Corollary 4.** *Information redundancy and supplement fusion* $((x)^{\overline{F}}, (x)^F)$ *is the* $(\alpha^F, \alpha^{\overline{F}})$-*information equivalence class* $[[x]^{\overline{F}}, [x]^F]$.

The following Propositions 4–6 are obtained directly by Theorems 9 and 10 and Corollary 4:

**Proposition 4.** *Information redundancy fusion* $(x)^{\overline{F}}$ *and* $\alpha^F$-*information equivalence class* $[x]^{\overline{F}}$ *are two equivalent concepts.*

**Proposition 5.** *Information supplement fusion* $(x)^F$ *and* $\alpha^{\overline{F}}$-*information equivalence class* $[x]^F$ *are two equivalent concepts.*

**Proposition 6.** *Information redundancy and supplement fusion* $((x)^{\overline{F}}, (x)^F)$ *and the* $(\alpha^F, \alpha^{\overline{F}})$-*information equivalence class* $[[x]^{\overline{F}}, [x]^F]$ *are two equivalent concepts.*

## 6. Application on Intelligent Acquisition of Information Equivalence Class in Information Fusion and Unknown Information Discovery

In order to be simple and easy to accept the conceptual and theoretical results given in Sections 3–5 of this paper, this section only gives the simple application of $\alpha^F$-information equivalence intelligence acquisition in information redundancy fusion and unknown information discovery.

Suppose that $x_1$, $x_2$, $x_3$, $x_4$, $x_5$, $x_6$, $x_7$ are PhD students enrolled in 2018, they will complete their PhD within four years; $x_1 \sim x_7$ come from different provinces in China; $x_1 \sim x_7$ constitutes information $(x)$:

$$
(x) = \{x_1, x_2, x_3, x_4, x_5, x_6, x_7\},
\tag{34}
$$

$\forall x_i \in (x)$ has the test scores of math, physics, computer, information technology: mathematics = $y_{1,i}$, physics = $y_{2,i}$, computer = $y_{3,i}$, information technology = $y_{4,i}$; $i = 1, 2, 3, 4, 5, 6, 7$. $A$ is the information value matrix generated by $(x)$:

$$
A = \begin{bmatrix}
87 & 93 & 79 & 97 \\
80 & 88 & 91 & 87 \\
74 & 83 & 92 & 77 \\
91 & 90 & 93 & 88 \\
96 & 73 & 82 & 91 \\
85 & 89 & 90 & 78 \\
91 & 91 & 70 & 85
\end{bmatrix},
\tag{35}
$$

where, for $x_i \in (x)$, the $j$ column $y_j$ in A are 4 scores: $y_{1,i}$, $y_{2,i}$, $y_{3,i}$, $y_{4,i}$, which constitutes the vector $y_j = (y_{1,i}, y_{2,i}, y_{3,i}, y_{4,i})^T$; $j \in (1, 2, 3, 4, 5, 6, 7)$, $i = 1, 2, 3, 4, 5, 6, 7$.

The math, physics, computer, and information technology are defined as attributes $\alpha_1$ = math, $\alpha_2$ = physics, $\alpha_3$ = computer, $\alpha_4$ = Information Technology respectively. $\alpha_1$, $\alpha_2$, $\alpha_3$, $\alpha_4$ constitutes the attribute set $\alpha$ of $(x)$:

$$
\alpha = \{\alpha_1, \alpha_2, \alpha_3, \alpha_4\}
\tag{36}
$$

Because each $x_i \in (x)$ has the attributes $\alpha_1, \alpha_2, \alpha_3$ and $\alpha_4$, the attribute $\alpha_i$ of $x_i \in (x)$ satisfies the attribute "conjunctive normal form" of Equation (15); that is,

$$\alpha_i = \alpha_1 \wedge \alpha_2 \wedge \alpha_3 \wedge \alpha_4 = \wedge_{t=1}^{4} \alpha_t. \tag{37}$$

If we want to know which one in $x_1 \sim x_7$ came from Shandong Province, China, we can add the attribute $\alpha_5 =$ Shandong Province to the attribute set $\alpha$, so $\alpha$ generates $\alpha^F$:

$$\alpha^F = \alpha \cup \{\alpha_5\} = \{\alpha_1, \alpha_2, \alpha_3, \alpha_4, \alpha_5\}. \tag{38}$$

We get the $(x)^{\overline{F}}$ with attribute set $\alpha^F$:

$$(x)^{\overline{F}} = (x) - \{x_3, x_5, x_6, x_7\} = \{x_1, x_2, x_4\}. \tag{39}$$

The attribute $\alpha_i$ of $\forall x_i \in (x)^{\overline{F}}$ satisfies the attribute "expansion of conjunctive normal form" of Equation (16), that is,

$$\begin{aligned}
\alpha_i &= (\alpha_1 \wedge \alpha_2 \wedge \alpha_3 \wedge \alpha_4) \wedge \alpha_5 \\
&= (\wedge_{t=1}^{4} \alpha_t) \wedge \alpha_5 \\
&= \wedge_{t=1}^{5} \alpha_t.
\end{aligned}$$

From Theorem 3, we can obtain that: $(x)^{\overline{F}}$ is the $\alpha^F$-information equivalence class $[x]^{\overline{F}}$ generated by $(x)$; from Theorem 10 and Proposition 4, we obtain that $(x)^{\overline{F}}$ is the information redundant fusion generated by $(x)$ by deleting $x_3, x_5, x_6, x_7$. The information value matrix $A$ generate the internal p-matrix $A^{\overline{F}}$:

$$A^{\overline{F}} = \begin{bmatrix} 87 & 93 & 79 & 97 \\ 80 & 88 & 91 & 87 \\ 91 & 90 & 93 & 88 \end{bmatrix}. \tag{40}$$

From Equations (35) and (40), we get that $A$ and $A^{\overline{F}}$ satisfy $A^{\overline{F}} \subseteq A$, or $A^{\overline{F}} \Rightarrow A$; Equations (35) and (40), Equations (34) and (39) satisfy the inner p-p-matrix reasoning respectively: *if* $A^{\overline{F}} \Rightarrow A$, *then* $(x)^{\overline{F}} \Rightarrow (x)$. Because the inner p-matrix reasoning condition is satisfied, $A^{\overline{F}} \Rightarrow A$, information redundancy fusion $(x)^{\overline{F}}$ is intelligently acquired in the information $(x)$; the students $x_1$, $x_2$, $x_4$ who come from the Shandong province are found in $x_1$-$x_7$.

From this simple example, we conclude that if the attribute $\alpha_5$ is added to the attribute set $\alpha$ of the information $(x)$, the redundant fusion $(x)^{\overline{F}}$ of unknown information is discovered intelligently from $(x)$; the unknown information $(x)^{\overline{F}}$ is hidden in $(x)$ before the attribute $\alpha_5$ is added to $\alpha$.

For $x_1, x_2, x_3, x_4, x_5, x_6, x_7$, we have conducted the survey of students from the provinces respectively, and the results of the survey are the same as those given in Equation (39).

## 7. Conclusions

For the p-set and its augmented matrix, some new conclusions are obtained from the analyses given in this paper as following:

1.  Under the condition that some attributes are added to the attribute set $\alpha$ of $(x)$, some information elements are deleted in $(x)$, so $(x)$ generate $(x)^{\overline{F}}$; that is, the boundary of $(x)$ shrinks inward to generate $(x)^{\overline{F}}$; by using the equivalence class concept in mathematics, we get that: $(x)^{\overline{F}}$ is the $\alpha^F$-information equivalence class $[x]^{\overline{F}}$ generated by $(x)$; the reasons are as following: the attribute set $\alpha^F$ for $\forall x_i, x_j, x_k \in [x]^{\overline{F}}$ satisfies the characteristics of the equivalence class: reflexivity, symmetry, and transitivity. Obviously, $(x)$ generates multiple $\alpha^F$-information equivalence classes $[x]_1^{\overline{F}}, [x]_2^{\overline{F}}, \cdots, [x]_n^{\overline{F}}$ under the condition of constantly supplementing attributes in $\alpha$; so $(x)$ continuously deletes the information element $x_i$ to get multiple information fusions: $(x)_1^{\overline{F}}, (x)_2^{\overline{F}}, \cdots, (x)_n^{\overline{F}}$;

2. each $(x)_i^{\overline{F}}$ is called information redundancy fusion, $i = 1, 2, \cdots, n$. we get a new concept of information fusion: information redundancy.

2. Under the condition that some attributes are deleted from attribute set $\alpha$ of $(x)$, some information elements are added in $(x)$, so $(x)$ generates $(x)^F$; that is, the boundary of $(x)$ expands outward to generate $(x)^F$; by using the equivalence class concept in mathematics, we get that: $(x)^F$ is the $\alpha^{\overline{F}}$-information equivalence class $[x]^F$ generated by $(x)$; the reasons are as following: the attribute set $\alpha^{\overline{F}}$ for $\forall x_i, x_j, x_k \in [x]^F$ satisfies the characteristics of the equivalence class: reflexivity, symmetry and transitivity. Obviously, $(x)$ generates multiple $\alpha^{\overline{F}}$-information equivalence classes $[x]_1^F, [x]_2^F, \cdots, [x]_n^F$ under the condition of continually deleting attributes in $\alpha$; so $(x)$ constantly supplement the information element $x_j$ to get multiple information fusions: $(x)_1^F, (x)_2^F, \cdots, (x)_n^F$; Each $(x)_j^F$ is called information supplementation fusion, $j = 1, 2, \cdots, n$. We get a new concept of information fusion: information supplementation fusion. Information redundancy fusion and information supplementation fusion exist in many application researches of information fusion.

The literature cited is [1–8,42,43] gives many excellent researches on multiple information fusions. By comparing with the literature [1–8,42,43]. The research given by the contributions of this article are as following: two new concepts of information fusion are presented by using mathematical methods to understand the concept and characteristics of information fusion: information redundancy fusion and Information supplementation fusion. The concept of information equivalence class is presented by using the new mathematical model: the p-set. Information equivalence class and information fusion are two equivalent concepts, which is an important theoretical conclusion. Information fusion intelligent acquisition method and intelligent acquisition algorithm are presented under the matrix reasoning conditions. The results given in the paper are all new.

**Author Contributions:** Conceptualization and formal analysis, S.L. and K.S.; writing—original draft preparation, Y.X. All authors have read and agreed to the published version of the manuscript.

**Funding:** This document is the results of the research project funded by National Natural Science Foundation of China (71663010), National Philosophy and Social Science Foundation of China (17BGL001), Shandong Provincial Natural Science Foundation of China (ZR2019MG015).

**Conflicts of Interest:** The authors declare there is no conflicts of interest regarding the publication of this paper.

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
