# Peer review of "Dynamic Boundary of P-Set and Intelligent Acquisition for Two Types of Information Fusion"

_computers, doi:10.3390/computers9010003_

Round 1

Reviewer 1 Report

The authors note down a lengthy mathematical proof which looks impressive , but is generally hard to follow and difficult to asses. 

I would recommend a journal with a more mathematical background.
The topic and content seems out of scope for this journal.

The introduction does not explain any clear practical application of P-sets or P augmented matrices to the reader, related to some field in computer science. I would expect some link is explained to a practical application in computer sciences, as aligned with the foci of this journal. Instead the whole paper seems one large mathematical proof, which is not the intention of this journal I believe. I do not judge the soundness of this proof, but because of this lacking introduction, the readers do not understand the relevance of the described problem. Please explain better if you think that this is problem relevant to practical computing problems.

The experimental validation of the proof seems very shallow. Only a very small dataset is used, and is not clear from the notation what is meant. e.g. x_1, x_n , each student is a column is this matrix?
Also section 5.1 is very short, I do not understand the significance of this subsection.

to summarize, There are some important things missing in this paper:
-A clear link of the problem to applied computing. What are some intended  practical use-cases? 

- A clear description how this paper compares to the state-of-the-art in the field. (and not just listing a series of references as in line 48)

- a more thorough practical validation of the solution with more elaborate test data. The validation should also show why the proposed solution is better than other existing methods.

Author Response

Thank you for your suggestions.

The introduction introduces the application of P-Set in the field of computer.

The introduction explains the relationship between P-Set dynamic boundary and two kinds of information fusion.

The purpose of very small data sets is to show the application of P-sets in information fusion and unknown information mining, regardless of the size of data. We will use larger data sets for further research.

We change the equations (35) and (40) in this paper to the conventional representation, that is, the row of matrix represents the student, and the column of matrix represents the course.

This article incorporates Section 6.1 into the previous section.

Reviewer 2 Report

This paper introduces theoretical and applied research on a new mathematical model of information fusion based on the concepts of P-sets and P-augmented matrices. My comments are as follows:

1- I do not think the whole concept is new, as it is claimed in the paper. I do not want to ignore the contribution of the paper, just removing the world 'new' might be an alternative. 

2- How the current draft differ from the following paper?

"Outer P-information law reasoning and its application in intelligent fusion and separating of information law." Microsystem Technologies 24, no. 10 (2018): 4389-4398.

3- The main contribution of the paper is not well described in the abstract. The authors need to rewrite the abstract by emphasizing the main contribution more clearly.

4- I highly encourage the authors to include some of the recently published papers in their introduction. In particular:

- "Estimation, inference and learning in nonlinear state-space models." PhD diss., PhD thesis, Texas A&M University, College Station, TX, 2019.

- Shuilian Xie., et. al.,  “Nonstationary Linear Discriminant Analysis,” 51st Asilomar Conference on Signals, Systems, and Computers (pp. 161-165), Pacific Grove, CA, 2017.

- "Finite-Horizon LQR Controller for Partially-Observed Boolean Dynamical Systems", Automatica, 95, p. 172-179, 2018.

- "Control of gene regulatory networks using Bayesian inverse reinforcement learning." IEEE/ACM Transactions on Computational Biology and Bioinformatics (TCBB) 16, no. 4 (2019): 1250-1261.

5- Adding a better diagram, consisting of more explanations, can be helpful. 

6- The format of some the references is not in standard form. These need to be fixed.

Author Response

Since the P-Set theory has been put forward for more than 10 years, the word "new" has been deleted in this paper. The difference from the literature "outer p-information law reasoning and its application in intelligent fusion and separating of information law" is that the P-Set proposed in this paper is composed of inner P-Set and outer P-Set. The previous paper is about the study of P-Set out of function and its information fusion and separation. Based on the dynamic boundary of P-Set, this paper focuses on the fusion of two kinds of information. It is a further research expansion based on the previous article. We rewrote the summary. Thank you very much for the relevant research literature provided by the review experts. On the basis of careful reading and understanding, we have made appropriate references in this paper. Thanks to the reviewers for pointing out the problems in the algorithm flowchart, we have added explanations to make the flowchart easier to read and understand. We revised the format of the references.

Round 2

Reviewer 1 Report

The improved introduction helps a lot for unfamiliar readers to understand the relevance of this topic in the field of computing. Also the flowchart clarifies things better.
However, the example data used to illustrate the new theorems, is very simple and does not convince the reader of the importance or usefullness of this new theory.
All in all, the paper presents the topic reasonably well.

Reviewer 2 Report

The paper is well- revised and in my opinion is ready for publication in the current form.